# DTLR-CS: Deep tensor low rank channel cross fusion neural network for reproductive cell segmentation

Xia Zhao[1]☯, Jiahui Wang[2]☯, Jing Wang[1], Jing Wang[1], Renyun Hong[1], Tao Shen[1], Yi Liu[2], Yuanjiao Liang◉[1] *

**1** Reproductive Medicine Center, Zhongda Hospital, Southeast University, Nanjing, Jiangsu Province, China, **2** School of Medicine, Southeast University, Nanjing, Jiangsu Province, China

☯ These authors contributed equally to this work.
* yuanjiao1965@126.com

**Data Availability Statement:** Our code is publicly available via GitHub (https://github.com/sdcly2023/DTLR-CS).

**Funding:** The work was supported by Special Fund Project for Health Science and Technology

## Abstract

In recent years, with the development of deep learning technology, deep neural networks have been widely used in the field of medical image segmentation. U-shaped Network(U-Net) is a segmentation network proposed for medical images based on full-convolution and is gradually becoming the most commonly used segmentation architecture in the medical field. The encoder of U-Net is mainly used to capture the context information in the image, which plays an important role in the performance of the semantic segmentation algorithm. However, it is unstable for U-Net with simple skip connection to perform unstably in global multi-scale modelling, and it is prone to semantic gaps in feature fusion. Inspired by this, in this work, we propose a Deep Tensor Low Rank Channel Cross Fusion Neural Network (DTLR-CS) to replace the simple skip connection in U-Net. To avoid space compression and to solve the high rank problem, we designed a tensor low-ranking module to generate a large number of low-rank tensors containing context features. To reduce semantic differences, we introduced a cross-fusion connection module, which consists of a channel cross-fusion sub-module and a feature connection sub-module. Based on the proposed network, experiments have shown that our network has accurate cell segmentation performance.

## Introduction

The two most used methods in assisted human reproduction today are artificial insemination (AI) [1, 2] and in vitro fertilization-embryo transfer (IVF-ET) [3, 4]. Artificial insemination is a reproductive technique that uses scientific techniques to implant the removed sperm into the uterus to achieve conception. However [5], its limitation is the uncontrollability after implantation into the mother. Therefore, with the advancement of technology, IVF-ET is created to assist human reproduction. IVF-ET is a technique in which the removed ovum and sperm are medically combined outside the body to form a germ cell, which is then developed into an embryo and implanted into the uterus for conception [6]. It offers help to infertility patients because of its technical integrity and high success rate, but it may also cause some pregnancy complications [7]. To reduce complications, advances in assisted human reproduction

Development of Nanjing Health Construction Commission (No. YKK22277) and Self-select Fund of Zhongda Hospital (No. ZDYYZXKT2019001) The funders had no role in study design, data collection and analysis, decision to publish, or preparation of the manuscript.

**Competing interests:** The authors have declared that no competing interests exist.

technology can be aided by cell image segmentation techniques [8] that detect germ cells as they are being formed. Cell segmentation [9] is the process of separating distinct sections of cell pictures based on attributes including texture, color, grayscale, and geometry. In one region, the aforementioned characteristics will be consistent or similar. However, in other regions, they will differ greatly. Numerous techniques, such as threshold segmentation [10], region expanding method [11], watershed algorithm [12], active contour segmentation [13], and deep learning-based algorithms [14], have been proposed in the extensive research on cell segmentation. Below is a description of a few of the most popular segmentation algorithms.

The Ostu [15], entropy [16], p-tile [17], and minimum error approaches are just a few examples of the many techniques available for choosing thresholds. The fundamental idea of the threshold [18] segmentation method is to categorize pixel data by setting distinct thresholds. Because of its straightforward implementation procedure and minimal computational effort, this method is frequently utilized in the first segmentation of blood cells. The theory of topological mathematical morphology serves as the foundation for the segmentation approach known as the watershed method [19]. The main idea is to think of the image as a topological landform in geodesy, where the gray value of each pixel corresponds to the elevation of a point. Uneven distribution of cell adhesion frequently occurs during the cell segmentation process, and the watershed technique is effective at resolving this issue. To prevent over-segmentation, it is typically essential to pre-process the image and combine sections. This is because when the watershed algorithm is used for image segmentation, it is simple to over-segment the image and the method is more sensitive to noise. Song et al. [20] proposed an image segmentation technique combining a watershed algorithm with a fuzzy C-mean clustering approach, and they ran tests on 39 images of three different histological kinds. The issue of overlapping cell pictures can also be effectively resolved by the active contour line model in addition to the watershed technique. In order to solve the issue of multiple objects overlapping, Ali et al. [21] proposed an active contour line model based on boundary and region information. However, due to the nonconvexity of the model, the model was sensitive to the initial position choice and may experience local extra or even divergence during the training process.

With the advent of deep neural networks, this technique has made great progress in the field of computer vision [22]. Deep neural networks have so far improved the detection and segmentation of cell pictures. As the pioneer of convolutional neural networks (CNN) [23] in image segmentation tasks, the full convolutional network (FCN) [24] lays the foundation of CNN in image segmentation. Similar to the FCN, U-Net [25] was proposed to solve the segmentation problem of medical images, which contains an encoding-decoding layer structure as well as a jump connection layer structure. In order to help with the supervision of correct cell segmentation, Chen et al. [26] suggested a deep contour-aware network with a multi-task learning framework. Regularization was applied to the network training process to further enhance the network's discriminative power. The nucleus and cell boundaries were separated using a deep neural network and a region-growing technique by Kumar et al [27]. To separate the nucleus from the background, Naylor et al. [28] employed a deep neural network. The projected probability map was then post-processed using a watershed technique. Cui et al. [29] created the FCN to predict cell nuclei and cell boundary information simultaneously. They then post-processed the segmented cell nuclei images by dividing a single image into multiple image blocks during the prediction process and then stitching them together. This method accurately segments a $1000 \times 1000$ size cell image in 5 seconds. In order to accurately segment overlapping nuclei, Simon et al. [30] developed a new CNN for segmenting cell nuclei. They started by measuring the distance between each nucleus boundary and the center. Kowal et al. [31] proposed a method combining the CNN and watershed transform to segment breast cancer cells. They used a CNN to semantically segment the pre-processed cell images and then

tried to separate the nuclei using the watershed segmentation method. Although all of the aforementioned techniques have produced improved outcomes, they all have certain drawbacks. These techniques don't pay attention to global information throughout the image segmentation process and solely concentrate on local semantic information, which makes it difficult to accurately segment cells.

In order to sufficiently reduce the semantic differences between codec features, we abandon the use of the skip connection in the U-Net structure and instead design a new module to replace it. Specifically, on the one hand, we propose a channel-based cross-fusion sub-module that cross-fuses contextual features from the dimension of channels. This module fuses the output features from different layers of encoders to achieve an adaptive scheme for reducing semantic differences through collaborative learning rather than independent connections. On the other hand, we present the feature connection sub-module for connecting the fused features of the encoder with the features of the decoder. To solve the above problem in constructing contextual information, we put forward with a tensor generation module to generate a low-rank tensor and apply it to the connection of the encoder and decoder. The basic idea is that the tensor generation module generates low-rank tensors in each of the three dimensions, feeds them into the channel cross-fusion sub-module for feature fusion, and finally sums their three-dimensional features. We embed the above two modules into the U-Net network. Our main contributions to this paper are presented as follows:

- We fuse contextual features with a cross-fusion sub-module in the channel dimension, which achieves semantic discrepancy reduction by fusing the output features of different layer encoders through collaborative learning rather than independent connections.

- To solve the problem of insufficient channel attention information when constructing contextual information, a low-rank tensor generation module is proposed and applied to the connection of the encoder and decoder. The three-dimensional low-rank tensor is generated separately by the tensor generation module and input to the channel cross-fusion sub-module for feature fusion, respectively.

- Extensive experimental analyses based on the dataset are conducted to evaluate the performance of the proposed framework against the benchmark consisting of state-of-the-art cell segmentation approaches.

The rest of the paper is organized as follows: Section II presents a review of U-Net, tensor low ranking, and transformer. Section III shows the model of our proposed algorithm and the specific structure of each sub-module. Section IV presents extensive experimental simulations, as well as a detailed discussion of the results. Finally, Section V draws conclusions.

## Related work

### U-Net

U-Net [25], which was initially proposed to address the cell wall segmentation problem, is one of the earliest and most well-known techniques for segmenting medical images. U-Net is a fully symmetric U-shaped structure that can be separated into two halves. The first section uses a normal CNN architecture to represent a systolic path. A ReLU activation unit, a maximum pooling layer, and two consecutive $3 \times 3$ convolutions make up each block of the systolic path. The uniqueness of U-Net lies in his extended path, where feature maps are upsampled using $2 \times 2$ convolution at each level before being cropped and stitched to the upsampled feature maps from the corresponding layers in the shrinkage path. An additional $1 \times 1$ convolution is used to decrease the feature map to the required number of channels and create the

segmented image after two consecutive 3 × 3 convolutions and ReLU activations. In addition, pixel features with little contextual information at their boundaries must be removed, necessitating network cropping of the feature map. More significantly, it disseminates contextual information throughout the network, enabling it to use context to separate items from more extensive overlapping regions. The energy function of U-Net is given by the following equation:

$$E = \sum w(x) \log \left( p_{k(x)}(x) \right) \tag{1}$$

$$p_k(x) = \frac{e^{a_k(x)}}{\sum_{k=1}^{k} e^{a_k(x)}} \tag{2}$$

where $p_k$ denotes the softmax function, which is applied to the output feature map of the network, and $a_k(x)$ denotes the activation function in the $k$th channel.

Compared with FCN, U-Net creatively implements one-to-one correspondence between the encoder module and decoder module, passes the low-level feature map to the high-level feature map part through the skip connection (SC) structure. It fuses the low-level feature map with the high-level feature map for processing, and these operations help U-Net achieve excellent performance in the field of medical image segmentation. However, U-Net has some obvious flaws. A simple fusion method of splicing the lower-level information with the higher-level information does not fully consider the association between the lower-level information and the higher-level information.

## Low rank and tensor

Due to image redundancy and self-similarity, there are frequently many regular geometric textures and detailed structural features locally, causing the image matrix to exhibit local low-rank properties [32]. The matrix rank minimization has a strong global constraint and can accurately depict two-dimensional sparsity [33]. However, as modern information technology has advanced, the acquired high-dimensional data have more complex structures, such as color images, video sequences, HSI and MRI data, and so on. Traditional data representations (vectors or matrices) are incapable of accurately capturing the essential structure of these data. Tensor [34] can better express the complex essential structure of higher-order data as a higher-order extension of vector (first order) and matrix (second order) representations (data greater than or equal to third order is called higher-order data). Tensor, as a high-dimensional extension of the matrix, provides an efficient way to represent the structural properties of higher-order data. There is no consensus on the most appropriate definition of tensor rank, so it is common practice to design convex or non-convex alternative optimization schemes based on different definitions.

## Transformer

Following the introduction of transformer in recent research for visual identification tasks to model remote dependencies, numerous transformer variants, including Swin-Transformer [35], DieT [36], and TiT [37], have shown remarkable success in natural image recognition tasks. Using the powerful representation capability of transformer, several works have attempted to replace or combine CNNs to achieve better results for medical image segmentation. The transformer is an attention-based model that was previously designed for sequence prediction [38]. The key component of transformer, self-attention (SA), models the correlation between all input tokens, allowing the Transformer to handle dependencies over time.

Although some of these works have produced satisfactory results, they typically rely on large-scale pre-training, making the use of these methods inconvenient [39, 40].

## Materials and methods

### Overview

The most used segmentation design in the medical industry is the U-Net network. The contextual information in the image is primarily captured by the encoder component of the U-Net network, which is crucial to the efficiency of the semantic segmentation method. Due to the diversity of the contextual information, the features must be represented by a tensor of high rank, which necessitates the use of numerous parameters, which can be quite expensive. We can extract contextual information using a tensor low-ranking module, which is inspired by tensor decomposition theory. The features produced by our encoder will be transmitted to a decoder at a higher tier than the original network when this module is used in a skip connection of a U-Net network. We do away with the straightforward skip connection and design a new cross-fusion connection module to cross-fuse the encoder features with the connected codec features, thereby reducing the differences between encoder and decoder features because jump connections are not always effective and sometimes even have negative effects, which we analyze are due to the differences between encoder and decoder features. A channel cross-fusion sub-module and a feature connection sub-module make up our cross-fusion connection module. Fig 1 shows the structure of our proposed DTLR-CS network.

### Tensor low ranking module

The output of the context fragments, which are made up of vectors of rank 1 in the three directions of height, breadth, and channel, is the goal of our tensor low-ranking module. Therefore, in order to extract the context information in the three directions, we need three feature

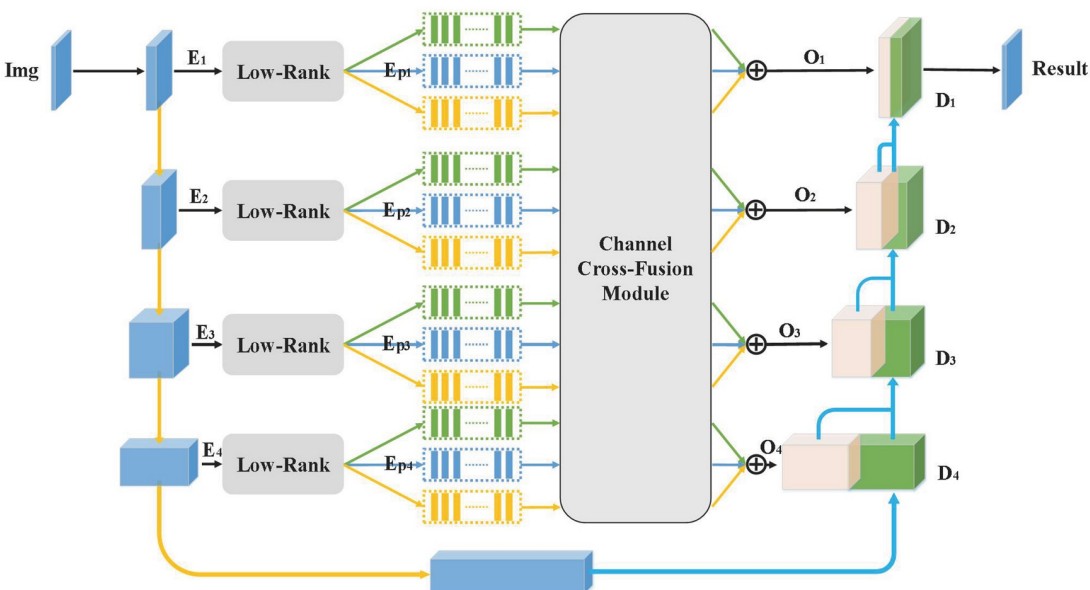

**Fig 1. The structure of our proposed network.** The original skip connection is replaced by a cross-fusion connection module after tensor low-ranking, which consists of a channel cross-fusion sub-module and a feature connection sub-module.

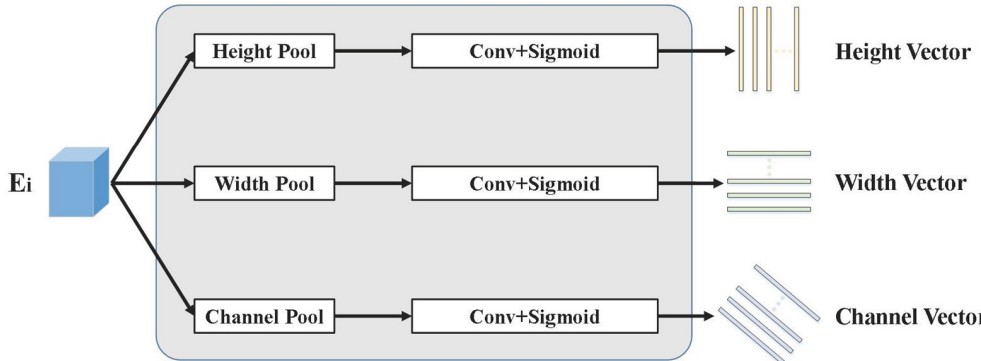

**Fig 2. The structure of the tensor low-ranking module.** This module consists of three parts: height feature generator, width feature generator, and channel feature generator.

extractors. A pooling layer, a convolutional layer, and an activation function make up our feature extractor. Our feature generator uses global average pooling as a means of obtaining contextual information because it is effective. These outputs must be nonlinear in order for each to reflect distinct information. In order to meet the requirements of attention, we employ the Sigmoid function, which rescales the features that pass through the convolutional layer into the range [0, 1]. We obtain 3L vectors of rank 1 by putting the input features through L different height feature generators, width feature generators, and channel feature generators, respectively. A component of the context fragment is present in each of these vectors. Fig 2 shows the structure of the tensor low-ranking module.

## Channel cross-fusion module

We suggest a three-step channel cross-fusion sub-module that combines contextual feature embedding, channel cross-fusion attention, and perceptron to solve the instability of skip connections. Fig 3 shows the structure of the channel cross-fusion. Incorporating contextual features. We first want to process these characteristics of distinct encoder layers by reconstructing them as patches sizes $P_s$, $\frac{P_s}{2}$, $\frac{P_s}{4}$, and $\frac{P_s}{8}$ for the output $E_{d,i}$(d = height, width, channel, i = 1, 2, 3, 4) of the tensor low-ranking module of four separate encoder layers. The goal is to make it possible for these features to map to the same cluster of features with low rankings on four scales. After that, $d$ will separate these $T_{d,i}$ and $T_{d,\sum_i}$ attributes into three categories.

The channel cross-attention sub-module then receives inputs from the $T_{d,i}$ and $T_{d,\sum_i}$, followed by a perceptron. The structure of the channel cross-attention sub-module is shown in Fig 4. It has 5 inputs, including $T_{d,i}$ and $T_{d,\sum_i}$. The outcome $O_i$ can be described as follow:

$$O_i = \sum_d (NCA_{d,i} + LP(Q_{d,i} + NCA_{d,i}))$$ (3)

The goal of this summing process, where $LP$ is the perceptron, is to retrieve the high-rank tensor together with the low-rank tensor, and $NCA_{d,i}$ can be written as:

$$NCA_{d,i} = (CA_{d,i}^1 + CA_{d,i}^2 + \cdots + CA_{d,i}^2)/N$$ (4)

where $N$ stands for the quantity of channel cross-notice submodules, and $CA_{d,i}$ is represented

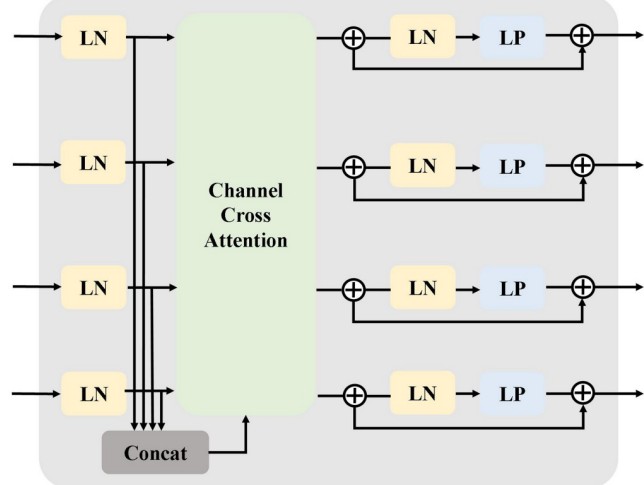

**Fig 3. The structure of the channel cross-fusion sub-module.** It consists of contextual feature embedding, channel cross-fusion attention, and perceptron.

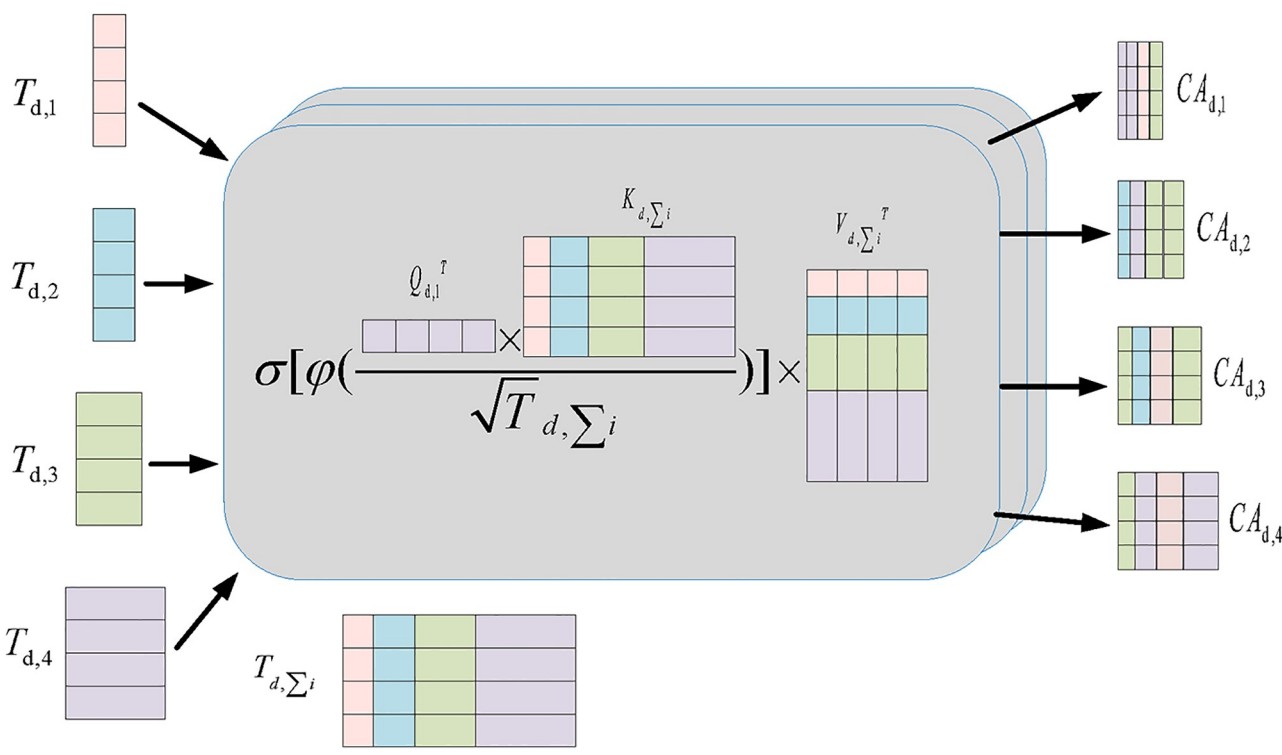

**Fig 4. Channel cross-fusion attention sub-module.**

by:

$$\mathrm{CA}_{d,i} = \sigma\left[\varphi\left(\frac{Q_{d,i}{}^T K_d}{\sqrt{\mathrm{T}_{d,\Sigma_i}}}\right)\right]V_d^T \tag{5}$$

where $\varphi(\cdot)$ and $\sigma(\cdot)$ denote the normalization operation and the activation function, respectively, and $Q_{d,i}$, $K_d$, and $V_d$ can be expressed as:

$$Q_{d,i} = \mathrm{T}_{d,i}W_{d,Q}$$

$$K_d = \mathrm{T}_{d,\Sigma i}W_{d,K} \tag{6}$$

$$V_d = \mathrm{T}_{d,\Sigma i}W_{d,V}$$

where $\mathrm{T}_{d,i}$ and $\mathrm{T}_{d,\sum_i}$ are the characteristics we provide into the channel cross-attention sub-module, and $W_{d,Q}$, $W_{d,K}$, and $W_{d,V}$ are the weights of the various inputs. In the derivation above, we simplified things by leaving out the normalizing layer from the equation.

## Feature connection module

In order to address the issue of incompatibility between the shallow feature set output from the encoder and the feature set output from the decoder, we create a feature connection sub-module to connect the fused features of the encoder and the decoder. Fig 5 shows the structure of feature connection sub-module. Prior to feature concatenation, the output $O_i$ of the channel cross-fusion sub-module must undergo upsampling and convolutional layer procedures. As the inputs for the feature concatenation sub-module, we use the computed oi and the decoder layer $i$ feature $D_i$. Then we conduct global average pooling on the computed $O_i$ and $D_i$, and input them to the two linear layers. By adding the two linear layer outputs, we connect $O_i$ and $D_i$, and then combine the total of those results with the computed oi to get the connected features.

## Loss fuction

The weighted cross-entropy (WCE) loss and the dice loss are the two halves of the loss function that we use in our method. Cross-entropy is frequently employed in classification tasks, and as image segmentation is a pixel-level classification, it frequently produces good results.

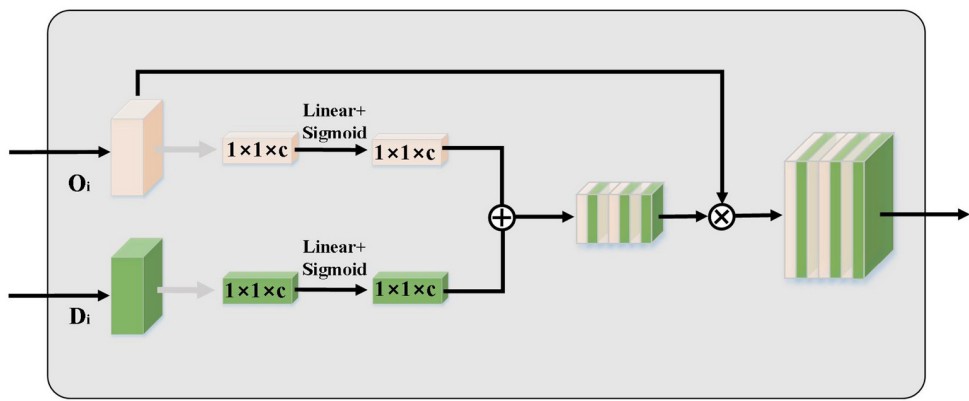

**Fig 5. Feature connection sub-module.**

The cross-entropy loss function is effective for the majority of semantic segmentation scenarios, but for tasks like cell segmentation where only two cases: foreground and background. It need to be separated and the number of foreground pixels is significantly less than the number of background pixels, the model may be heavily biased toward the background, leading to subpar segmentation outcomes. To address the issue of class imbalance, the weighted cross-entropy loss function augments the cross-entropy loss function with a weight parameter for positive examples of each class. The following equation can be used to express the weighted cross-entropy:

$$\text{WCE}(p, \hat{p}) = -(\beta p \log(\hat{p}) + (1 - p) \log(1 - \hat{p})) \tag{7}$$

Dice loss has been frequently employed in medical picture segmentation problems since it was initially presented in the article VNet [41]. The dice coefficient, which is frequently employed to determine how similar two collections are, serves as the foundation for dice loss and may be written as follows:

$$s = \frac{2 \mid X \cap Y \mid}{\mid X \mid + \mid Y \mid} \tag{8}$$

where $|X \cap Y|$ is the point at which sets $X$ and $Y$ intersect. $|X|$ and $|Y|$ stand for the number of elements in sets $X$ and $Y$, respectively. Because the denominator is used more than once to determine the common elements between $X$ and $Y$, the numerator coefficient is 2. The following equation can be used to describe dice loss:

$$\text{Dice} = 1 - \frac{2 \mid X \cap Y \mid}{\mid X \mid + \mid Y \mid} \tag{9}$$

where $X$ and $Y$ can denote the labeled image and the original image, respectively. Dice Loss is applicable to the case of extremely unbalanced class distribution, and is therefore suitable for segmentation of cellular images. The loss function of our method can be expressed as follows:

$$L = \alpha \cdot \text{WCE} + (1 - \alpha) \cdot \text{Dice} \tag{10}$$

## Experiment

### Datasets

The dataset used in this paper is a self-constructed dataset, utilizing reproductive cell images from the Reproductive Medicine Center of Zhongda Hospital affiliated with Southeast University in 2018 to train and evaluate our method. The germ cells that we selected are taken from infertile patients who underwent in vitro fertilization and embryo transfer. The number of ovum obtained is usually greater than or equal to eight, without limiting the cause of infertility or age. The equipment we have used is the Timelapse incubator by ASTEC in Japan and the model that we choose is the CCM-iBIS.

In order to ensure the credibility and effectiveness of the study, we have introduced certain data restrictions during the process of data collection. Firstly, the dataset exclusively comprises 2018 germ cell images, maintaining a consistent timeframe to reduce potential variations due to changes in practices, equipment, or patient demographics. Secondly, the dataset encompasses diverse germ cells without restricting the cause of infertility or patient age, capturing a wide range of real-world scenarios to enhance result generalizability and representativeness. Furthermore, a significant data limitation involves the minimum ovum count per patient,

specifically cases with eight or more ovum acquired. This stipulation ensures comprehensive germ cell development views per data point, reducing outlier impact and enhancing statistical analysis robustness. These restrictions establish a well-defined scope for the study, facilitating a focused and comprehensive analysis of germ cell behavior and development within the context of infertility and IVF procedures.

## Implementation and processing time

We use a 48 GB RAM NVIDIA A40 GPU card to implement our model in PyTorch [42]. To prevent over-fitting, we use flipping for data enhancement. We train all our models for 40 epochs on 1 GPU with a batch size of 4, and our initial learning rate is set to 0.001. We use the Adam optimiser to optimise our network. The Adam optimiser is simple to implement and computationally efficient. In addition, the update of the parameters using the Adam optimiser is not affected by the scaling transformation of the gradient, while the learning rate can be automatically adjusted, making it suitable for scenarios with large scale data and parameters. Training the model with the Deep Tensor Low Rank Channel Cross Fusion Neural Network took the shortest run time with the lowest trainable number of parameters improvement in the model performance.

## Compared methods

In order to analyze the overall performance of our method, we compare it with other advanced algorithms. We compare our method with U-Net, UNet++, Attention U-Net and TransUNet-based Transformer, whose original settings we used in our experiments.

- *U-Net* [25]. U-Net is a symmetrical encoder-decoder structure that incorporates a jump connection between the encoder and decoder, allowing the network to better fuse features at different scales. U-Net has the advantage of being flexible and simple and can achieve good segmentation results with relatively small sample datasets. Therefore, U-Net has been widely used in medical image segmentation.

- *UNet++* [43]. UNet++ is a variant of the U-Net network. Unlike U-Net networks where encoders and decoders are simply connected, a series of nested and dense jump connections narrow the information gap between coders and decoders in UNet++ networks. The network enable the richer fusion of low-dimensional information with higher-dimensional information, and thus extracting the hidden information of the original samples more effectively. By redesigning the jump connection structure, UNet++ model incorporate a dense connection strategy into the traditional U-Net network framework.

- *Attention U-Net* [44]. Attention U-Net uses a spatial domain-based attention mechanism to improve the performance of U-Net networks. Attention U-Net incorporates an attention module in the encoder and decoder. The addition of the attention mechanism to the encoder and decoder allows for more targeted segmentation of images. Attention U-Net can help U-Net to learn the interrelationships between multiple content modalities and represent these information better.

- *TransUNet* [45]. TransUNet is the first U-Net to apply the transformer to medical image segmentation. TransUNet serializes the down-sampled images directly from the encoder and applies them to the original transformer module in NLP for training.

### Evaluation metrics

To evaluate the effectiveness of the algorithm, this paper used Dice, mean intersection over union (MIoU), precision, and recall as evaluation metrics.

- *Dice* [46]. The dice similarity coefficient (DSC) is used as the evaluation criterion for the segmentation process. Dice is an aggregated similarity metric. The metric is usually used to calculate the similarity of two samples, with values ranging from 0 to 1, with the best segmentation result being 1 and the worst being 0. The formula for calculating Dice is as follows:

$$\text{Dice} = \frac{2\text{TP}}{2\text{TP} + \text{FP} + \text{FN}} \tag{11}$$

- *MIoU* [47]. Mean Intersection over Union (MIoU) as a standard metric for semantic segmentation. MIoU is used to calculate the ratio of the intersection and concatenation of the predicted and true results of the network, and then find the average value. The calculation formula is as follows:

$$\text{MIoU} = \frac{\text{TP}}{\text{TP} + \text{FP} + \text{FN}} \tag{12}$$

- *Precision* [48]. Precision is a metric widely used in the field of information retrieval and statistical classification. The metric represents the ratio of samples predicted to be correct to those predicted to be correct and is used to assess the quality of the results:

$$\text{Precision} \ = \frac{TP}{TP + FP} \tag{13}$$

- *Recall* [49]. Recall is the ratio of samples predicted to be correct to the number of positive samples in the sample. The metric is also used to evaluate the quality of the results. The calculation follows:

$$\text{Recall} \ = \frac{TP}{TP + FN} \tag{14}$$

where TP denotes the number of pixels predicted to be correctly classified as a target class, the number of predicted pixel sites that are incorrectly classified as a target category is referred to as FP, the number of predicted pixel points incorrectly classified as a non-target category is referred to as FN.

## Results and discussion

### Quantitative analysis

Fig 6 presents the results of cell segmentation using the method proposed in this paper. As can be seen from Fig 6, the method is able to accurately separate the cells from the background, achieving better results during the cell segmentation test.

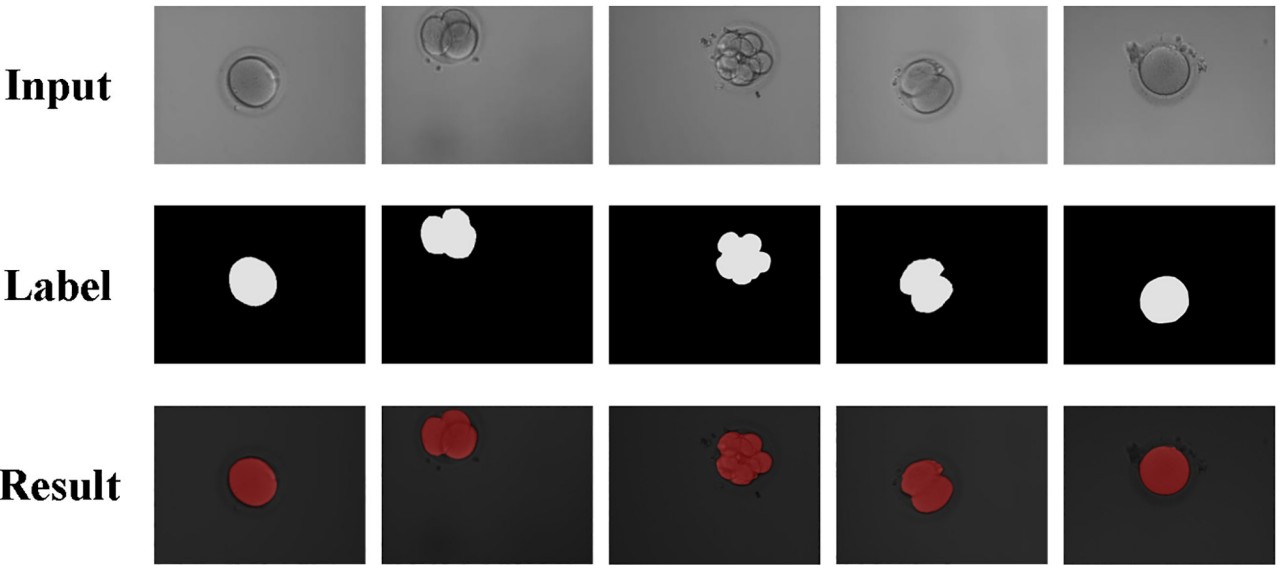

**Fig 6. Test results of the method.**

From Fig 7, we can see that as the number of training rounds increased, the precision continued to rise and then remained flat. At the same time, the loss rate continued to decrease, and then the curve showed small fluctuations before finally levelling off. As a result, our method has achieved good performance in image segmentation.

Fig 8 shows the Dice, MIoU, precision, and recall metrics of our method. After 40 rounds of epoch, the Dice metric is 98.58%, the MIoU metric is 97.23%, the precision metric is 98.47%, and the recall metric is 98.69%. This shows that our model achieves relatively accurate results for cell image segmentation. In order to analyse the overall performance of our method, we compared it with other algorithms. We compared our method with U-Net, UNet++, Attention U-Net and TransUNet. Table 1 showed the results of the comparison of Dice and MIoU metrics between the different algorithms. As seen in Table 1, our method improved over the

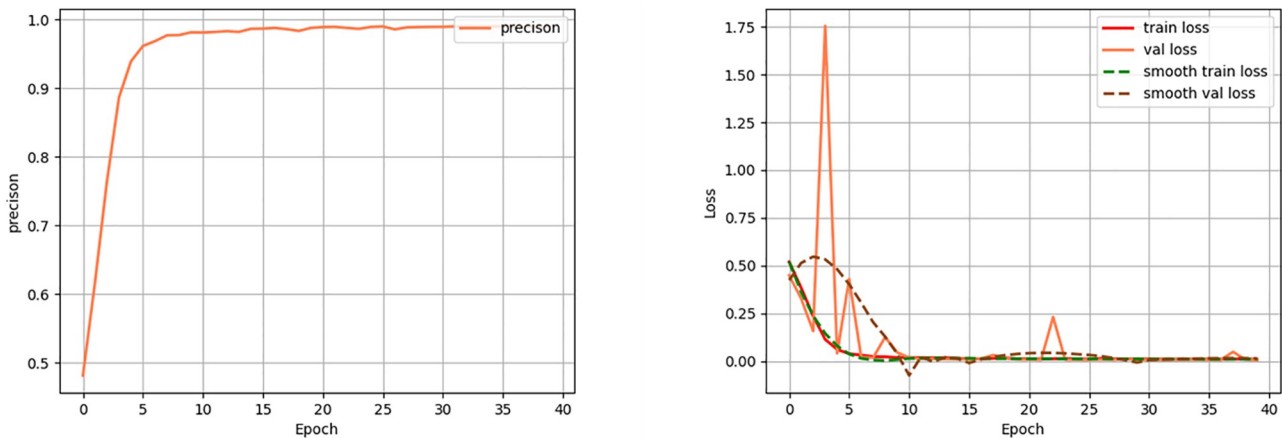

**Fig 7. Precision metric results and curves and loss values for 40 epoch rounds and loss function transformation curves.**



**Fig 8. Performance metrics of the method.**

other methods in terms of Dice and MIoU evaluation metrics. The Dice and MIoU of the method proposed in this paper are 98.58% and 97.23% respectively. Compared with the U-Net method, our proposed method has the 6.24% performance improvement in Dice and a 4.42% performance improvement in MIoU. Compared to the UNet++ method, the Dice of our method improved by 5.51% and the MIoU improved by 7.01%. Attention U-Net and TransU-Net achieved worse segmentation results with the same number of iterations. Therefore, the method proposed in this paper has better segmentation performance.

## Ablation study

In order to verify the effectiveness of the various modules in the proposed method, ablation study is conducted and five different networks were trained for comparative analysis. We choose U-Net as the first model because it is a common performance baseline for image segmentation. The second model introduced the CCF module based on the first model to address the instability of jump connections. The third model introduces the FC module based on the first model to solve the problem of incompatibility between the shallow feature set output from the encoder and the feature set output from the decoder. The fourth model adds the CCF module and the FC module to the U-Net network together. The fifth model introduces the LR module based on the fourth model, which is the model proposed in this paper. The above models were trained separately and the results are shown in Table 2. As can be seen from Table 2, compared to the traditional U-Net model, Dice and MIoU improve with the addition of either model.

The channel cross-fusion module fuses the output features of different layers of encoders and replaces independent connections with collaborative learning to reduce semantic differences, with a small improvement in the Dice metric. The feature connection module is used to connect the fused features of the encoder to the features of the decoder. If the channel cross-fusion module and the feature connection module are added, the Dice metric is improved relatively significantly, indicating that better cell segmentation can be achieved. The purpose of

**Table 1. Comparison of performance with other methods.** Our method is compared with Unet, Unet++, Attention Unet and TransUnet, respectively, and the best results are bolded.

| Method | Dice(%) | MIoU(%) |
|:---:|:---:|:---:|
| U-Net | 92.34 | 92.81 |
| UNet++ | 93.07 | 90.22 |
| Attention U-Net | 89.66 | 88.34 |
| TransUNet | 88.4 | 80.4 |
| Our | **98.58** | **97.23** |

**Table 2. Ablation study.** Our proposed method is Baseline+ LR+CCF+FC, and the best results are shown in bold.

| Method | Dice(%) | MIoU(%) |
|---|---|---|
| Baseline(U-Net) | 92.34 | 92.81 |
| Baseline+CCF | 93.25 | 89.23 |
| Baseline+FC | 92.55 | 90.02 |
| Baseline+CCF+FC | 96.32 | 92.65 |
| Baseline+LR+CCF+FC | **98.58** | **97.23** |

the tensor low-ranking module is aimed to get context fragments that can be obtained in multiple dimensions of space and channels. It solves not only the previous problem of feature compression but also the high-ranking difficulty. As shown in Table 2, both Dice and MIou metrics are substantially improved when the channel cross-fusion module, the feature connection module and the tensor low ranking module are added to the U-Net network at the same time, making this combination achieve the best performance. The proposed model has higher segmentation accuracy when dealing with cases where the cell background boundary is not obvious.

## Conclusion

In this paper, we propose a segmentation model based on U-Net that can efficiently achieve contextual information and reduce semantic differences, which obtains the contextual information contained in the high-rank tensor by low-ranking the tensor. Then the model passes the spatial information of the shallow layers to the decoder by fusing features from between different layers of the encoder. Finally, it connects the encoder and decoder by a cross-fusion connection module in order to recover the full spatial resolution. Experimental results show that our method is more competitive than existing segmentation methods.

## Author Contributions

**Data curation:** Xia Zhao, Jiahui Wang, Jing Wang, Renyun Hong, Yuanjiao Liang.

**Formal analysis:** Xia Zhao, Jiahui Wang, Jing Wang, Renyun Hong, Yuanjiao Liang.

**Methodology:** Tao Shen.

**Writing – original draft:** Jing Wang, Yi Liu.

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
