## [Decision Letter · Decision Letter 0]

2 Aug 2023

PONE-D-23-13899DTLR-CS: Deep Tensor Low Rank Channel Cross Fusion Neural Network for Reproductive Cell SegmentationPLOS ONE

Dear Dr. Liang,

Thank you for submitting your manuscript to PLOS ONE. After careful consideration, we feel that it has merit but does not fully meet PLOS ONE’s publication criteria as it currently stands. Therefore, we invite you to submit a revised version of the manuscript that addresses the points raised during the review process.

We look forward to receiving your revised manuscript.

Kind regards,

Xiaowei Li

Academic Editor

PLOS ONE

Journal Requirements:

2. Please share your code publicly, and provide a link in the Data availability statement.

3. We note that you have stated that you will provide repository information for your data at acceptance. Should your manuscript be accepted for publication, we will hold it until you provide the relevant accession numbers or DOIs necessary to access your data. If you wish to make changes to your Data Availability statement, please describe these changes in your cover letter and we will update your Data Availability statement to reflect the information you provide

4. Please remove your figures from within your manuscript file, leaving only the individual TIFF/EPS image files, uploaded separately. These will be automatically included in the reviewers’ PDF.

Reviewers' comments:

Reviewer's Responses to Questions

**Comments to the Author**

1. Is the manuscript technically sound, and do the data support the conclusions?

Reviewer #1: Yes

Reviewer #2: Partly

2. Has the statistical analysis been performed appropriately and rigorously? 

Reviewer #1: I Don't Know

Reviewer #2: Yes

3. Have the authors made all data underlying the findings in their manuscript fully available?

Reviewer #1: No

Reviewer #2: Yes

4. Is the manuscript presented in an intelligible fashion and written in standard English?

Reviewer #1: Yes

Reviewer #2: Yes

5. Review Comments to the Author

Reviewer #1: Thank you for the article. Kindly explain regarding the data restrictions in this project.

Great introduction in each area. Kindly transform the bullet points into paragraph format and consider to write the sub heading according to standard scientific paper sub heading.

Reviewer #2: In the work "DTLR-CS: Deep Tensor Low Rank Channel Cross Fusion Neural Network for Reproductive Cell Segmentation" Yuanjiao Liang and the co-authors have proposed a DeepTensor Low Rank Channel Cross Fusion Neural Network (DTLR-CS) to replace the simple skip connection in U-Net. They have designed a tensor low-ranking module to generate a large number of low-rank tensors containing context features and introduced a cross-fusion connection module, which consists of a channel cross-fusion sub-module and a feature connection sub-module. They have taken 2018 germ cell images to train and evaluate their method. My comments are stated below.

1. The Source of the dataset taken should be clearly mentioned in detail in the sub-section “Datasets”.

2. The authors have tried to identify the gaps or flaws in the related works done in the “Introduction” section. It is suggested to cite more references for the statements furnished there.

3. The Figure 4 and Figure 8 should be reconstructed to increase its visibility.

4. It is suggested to include a subsection for Implementation and processing time of the proposed method.

5. Please rewrite the “Conclusion” section to make it succinct.

6. PLOS authors have the option to publish the peer review history of their article (what does this mean?). If published, this will include your full peer review and any attached files.

Reviewer #1: No

Reviewer #2: No

---

## [Author Response · Author response to Decision Letter 0]

6 Sep 2023

Thanks to the reviewers for their valuable comments, which were very helpful to our paper. We have studied reviewer’s comments carefully and have tried our best to revise our manuscript according to the comments, in particular, the datasets and figures have been improved, relevant working sections have been added, and the language has been embellished.

The main changes are as follows:

1) We have reorganized the presentation of the datasets, explaining the dataset source and data restriction.

2) We corrected mistakes in grammar, spelling, formulas and figures in the article, and added some required references.

3) We have reworked the conclusion section to make it more succinct and easier to read. 

Please see "Response to Reviews" for more details

---

## [Decision Letter · Decision Letter 1]

8 Nov 2023

DTLR-CS: Deep Tensor Low Rank Channel Cross Fusion Neural Network for Reproductive Cell Segmentation

PONE-D-23-13899R1

Dear Dr. Liang,

We’re pleased to inform you that your manuscript has been judged scientifically suitable for publication and will be formally accepted for publication once it meets all outstanding technical requirements.

Kind regards,

Xiaowei Li

Academic Editor

PLOS ONE

Additional Editor Comments (optional):

Reviewers' comments:

Reviewer's Responses to Questions

**Comments to the Author**

1. If the authors have adequately addressed your comments raised in a previous round of review and you feel that this manuscript is now acceptable for publication, you may indicate that here to bypass the “Comments to the Author” section, enter your conflict of interest statement in the “Confidential to Editor” section, and submit your "Accept" recommendation.

Reviewer #3: All comments have been addressed

Reviewer #4: All comments have been addressed

2. Is the manuscript technically sound, and do the data support the conclusions?

Reviewer #3: Yes

Reviewer #4: Yes

3. Has the statistical analysis been performed appropriately and rigorously? 

Reviewer #3: Yes

Reviewer #4: Yes

4. Have the authors made all data underlying the findings in their manuscript fully available?

Reviewer #3: Yes

Reviewer #4: Yes

5. Is the manuscript presented in an intelligible fashion and written in standard English?

Reviewer #3: Yes

Reviewer #4: Yes

6. Review Comments to the Author

Reviewer #3: The paper is excellent. It's perfect now. Every proposition is fulfilled. It is a great pleasure reading it. It represents a true contribution to our scientific community. It is interdisciplinary!!!!Publish it!!!!

Reviewer #4: It has been observed that your work is ready for publication as a result of the changes you have made for the criticisms deemed necessary in your work.

7. PLOS authors have the option to publish the peer review history of their article (what does this mean?). If published, this will include your full peer review and any attached files.

Reviewer #3: No

Reviewer #4: No

---

## [Editor Report · Acceptance letter]

16 Nov 2023

PONE-D-23-13899R1 

DTLR-CS: Deep Tensor Low Rank Channel Cross Fusion Neural Network for Reproductive Cell Segmentation 

Dear Dr. liang:

I'm pleased to inform you that your manuscript has been deemed suitable for publication in PLOS ONE. Congratulations! Your manuscript is now with our production department. 

Kind regards, 

on behalf of

Dr. Xiaowei Li 

Academic Editor

PLOS ONE